# Enhancing programmatic scale-up: Applying the consolidated framework for implementation research to evaluate decentralized drug-resistant tuberculosis services in Southern Nigeria

Ngozi Murphy-Okpala[1], Chinwe Eze[1], Edmund Ndudi Ossai[2,3], Chibuike Innocent Agu[3]*, Ifeyinwa Ezenwosu[4], Charles Nwafor[1], Ngozi Ekeke[1], Anthony Meka[1], Sode Matiku[5], Beatrice Kirubi[6], Okechukwu Ezeakile[1], Martin Njoku[1], Francis S. Iyama[1], Jacob Creswell[6], Victor Babawale[7], Chukwuma Anyaike[7], Joseph Chukwu[1]

1 Programs Department, RedAid Nigeria, Enugu, Nigeria, 2 Department of Community Medicine, Ebonyi State University, Abakaliki, Nigeria, 3 Department of Community Medicine, Alex-Ekwueme Federal University Ndufu-Alike, Abakaliki, Ebonyi State, Nigeria, 4 Department of Community Medicine, University of Nigeria Teaching Hospital, Nsukka, Nigeria, 5 New Dimension Consulting, Dar es Salaam, Tanzania, 6 STOP TB Partnership, Geneva, Switzerland, 7 National TB, Leprosy and Buruli ulcer Control Program, Abuja, Nigeria

* aguchibuike14@yahoo.com

## Abstract

### Background

Decentralization of Drug-Resistant Tuberculosis (DR-TB) services using multilevel interventions was piloted in Akwa-Ibom and Oyo States of Nigeria, which had high rates of pre-treatment loss-to-follow-up in 2021. The varying outcomes of the intervention strategies necessitated understanding what worked well and why. This study aimed to identify enablers and barriers shaping the implementation of decentralized DR-TB services in these states and provide actionable strategies for programmatic scale-up.

### Methods

Semi-structured key informant interviews were conducted with 40 stakeholders involved in the pilot implementation of decentralized DR-TB services in southern Nigeria. Interviews were audio-recorded, transcribed verbatim, and template analysis done using NVivo statistical software, adapting the Consolidated Framework for Implementation Research (CFIR) constructs and sub-constructs as priori codes for data synthesis and analysis.

### Results

The study identified enablers and barriers across the five CFIR domains. Intervention characteristics facilitators stemmed from its relative advantage and design enabling faster notification of diagnosed DR-TB patients, enhanced patient tracking, ease of conducting

**Data availability statement:** All relevant data are available in the Supporting information file (S3 TRANSCRIPT in S3 File).

**Funding:** The research leading to these results was funded by the Stop TB Partnership under TB REACH Wave 9 grant number: STBP/TBREACH/GSA/W9-9833. However, the views expressed do not necessarily reflect the views of Stop TB Partnership, but belong solely to the authors. The funders had no role in study design, data collection and analysis, decision to publish, or preparation of the manuscript" in the manuscript.

**Competing interests:** The authors have declared that no competing interests exist.

**Abbreviations:** DR-TB, Drug-Resistant Tuberculosis; MDR-TB, Multi-Drug Resistant Tuberculosis; CFIR, Consolidated Framework for Implementation Research; TB, Tuberculosis; NTBLCP, Nigerian Tuberculosis and Leprosy Control Program; PMDT, Programmatic Management of Drug-resistant Tuberculosis; TBLS, Tuberculosis and Leprosy Supervisor; PTLTFU, Pre-treatment loss to follow up; CBOs, Community-based organizations; USSD, Unstructured Supplementary Service Data; VLO, Volunteer liaison officer; LGAs, Local Government Areas; KIIs, key informant interviews; DOTS, Directly observed treatment, short-course; FP, Focal Person; WHO, World Health Organization; FMOH, Federal Ministry of Health.

baseline investigations, bringing treatment closer by reducing transportation challenges, unique role of the Volunteer Liaison Officer, utility of WhatsApp platform, private sector engagement, and DR-TB survivors as peer counsellors. Critical incidents such as the removal of fuel subsidies and economic difficulties; and inner setting factors like existing infrastructure and health workforce, culture, available resources and tension for change from high pre-treatment loss to follow-up, and intervention's alignment with workflow; all facilitated implementation. Barriers identified included: challenges with verbal autopsy, low adoption of Unstructured Supplementary Service Data (USSD) innovation, pricing of baseline tests, poor power supply, inadequate laboratory facilities and insufficient DR-TB expertise in rural areas, and fear of TB infection among health workers.

## Conclusion

The findings demonstrate the ease of implementing decentralized DR-TB services and their advantages over a centralized approach. Key enablers centered on innovation and individual characteristics, and inner setting dynamics within the TB program. There were more facilitators than barriers, with most barriers being modifiable despite some outer setting factors like fiscal policy and geographic access. These insights can guide nationwide adoption and scale-up of decentralized DR-TB services in Nigeria and similar settings in low-and middle-income countries.

## Trial Registration

Pan African Clinical Trial Registry PACTR202309676675265

## Background

Globally, of the estimated 410,000 people who developed Multi-Drug Resistant Tuberculosis (MDR-TB) in 2022, only 42.8% of them were diagnosed and started on treatment, with a treatment success rate of 63% [1]. Nigeria is among the thirty high tuberculosis (TB) burden countries that accounted for 87% of the global TB cases in 2022 and bears the highest burden of TB in Africa. Nigeria is included in the three global lists of high burden countries for TB, TB/HIV and MDR-TB, and is among the top ten countries that accounted for the global gap in estimated number of people who developed MDR/RR-TB and the number of people diagnosed and enrolled for treatment in 2022 [1].

Drug-resistant TB (DR-TB) services in Nigeria have traditionally been centralized with upwards referral of all DR-TB patients for management at the various state capitals led by the State-level TB team. Moreover, previous injectable-containing treatment regimens encouraged facility-based care. The first attempt at decentralizing DR-TB treatment was in 2013 when the National Tuberculosis and Leprosy Control Program (NTBLCP) adopted ambulatory care for Programmatic Management of DR-TB (PMDT). With this, services for treatment initiation and much of the intensive phase was mostly received as in-patients while the continuation phase was completed in the community under the care of the local government TB supervisors (TBLSs).

Since the advent of effective all-oral shorter DR-TB medications in 2018, the NTBLCP aligned with the global consensus to phase out injectable-containing treatment regimens and transition PMDT from being facility-based to community based for better access to patients and treatment outcome. This policy shift notwithstanding, all primary program structures

related to activities for treatment initiation such as conducting baseline investigations, logistics for medication access and patient's clinical review have largely remained centralized. There has been a consistent DR-TB diagnosis-enrollment gap in Nigeria, with a 3-year (2020–2022) national average enrollment rate of 76% among annually notified DR-TB cases; with lower proportions at sub-national levels [2]. Hence, despite the DR-TB under-diagnosis, about a quarter of those diagnosed are not enrolled on appropriate treatment, thereby contributing to continued transmission of the drug-resistant strain of TB bacilli.

Under the TB REACH Wave 9 project grant titled, *'Catalyzing improvements in DR-TB care in Nigeria: A sustainable patient-centered approach',* RedAid Nigeria applied a patient care pathway approach to design multilevel interventions to decentralize DR-TB services with the aim of improving linkage-to-care to reduce pre-treatment loss to follow up (PTLTFU). The pilot project was implemented over a 15-month period. Evidently, management of DR-TB using a decentralized approach is more effective than centralization as it improves treatment success rate, lowers PTLTFU and reduces the time to treatment initiation [3,4]. Furthermore, decentralization encourages task shifting thus empowering community-based health workers and reducing patient overload among the few specialized health workers. It has also been found to be cost-effective especially in low-resource settings [5].

This study is a systematic assessment of factors influencing the interventions' implementation using the Consolidated Framework for Implementation Research (CFIR) [6] through the lens of the key stakeholders who were directly involved in implementation. The CFIR is a widely used and well-operationalized framework that is designed to identify barriers and facilitators during different phases of activity implementation in local settings [7]. Furthermore, it offers a set of standardized and comprehensive implementation constructs, which account for characteristics that may influence implementation and thus play an essential role in whether (or not) interventions are successful [7,8]. Using the CFIR, this study aimed to describe the implementation experience, and identify factors that hindered or facilitated the implementation of different interventions to decentralize DR-TB services. This will provide comprehensive scientific guidance for policymakers and healthcare workers to improve future efforts to scale-up decentralized DR-TB services.

## Methods

### Conceptual framework

The CFIR consists of five domains: (a) Innovation (the "thing" being implemented); (b) Inner setting (the setting in which the intervention is implemented); (c) Outer setting (the setting in which the Inner Setting exists); (d) Individuals (roles and characteristics of individuals); and (e) Implementation process (activities and strategies used to implement the interventions). An established determinants framework, the CFIR constructs and sub-constructs under these five domains reflect the evidence base of factors most likely to affect the implementation of interventions [6,7]. CFIR is therefore appropriate for assessing the factors that facilitate and impede decentralization of DR-TB services, enabling researchers to evaluate these factors and their interrelationships.

**Study design.**  This was a cross-sectional study using qualitative data collection methods.

**Study setting.**  Study was conducted in two southern states (Akwa-Ibom and Oyo) of Nigeria where the TB REACH Wave 9 project was implemented. The inhabitants of both states are mostly rural dwellers with some urban and peri-urban centres, but also have urban slums with overcrowded and poorly ventilated houses [9]. There are a total of 671 and 538 DOTS facilities in the project states (Oyo and Akwa-Ibom respectively). The project states have 28 GeneXpert machines (Akwa-Ibom: 13, Oyo: 15). Oyo State has two treatment centres

and a (south-west zonal) reference laboratory, unlike Akwa-Ibom State which only has one DR-TB treatment centre and uses either the south-east or south-south zonal reference lab located at State Specialist Hospital, Amachara, Abia State and University of Port-Harcourt Teaching Hospital, Rivers State respectively.

## Description of the interventions

This study was conducted retrospectively to evaluate the implementation of the TB REACH Wave 9 project grant titled, *"Catalyzing improvements in DR-TB care in Nigeria: A Sustainable Patient-centered approach"*. The project aimed to decentralize DR-TB services through multi-component interventions. The interventions implemented were: Devolving DR-TB diagnosis notification to include the local government TB supervisors (TBLS) using WhatsApp; Tracking of patients by supporting the Global Fund-funded community-based organizations (CBOs) through output-based financial incentives; Counselling of patients with the introduction of a structured pre-treatment counselling checklist and team counselling with DR-TB survivors; Decentralization of baseline investigations to (pre-qualified) peripheral laboratories; Provision of transport support to patients for baseline investigations (this was not directly implemented by the project as the Global Fund started funding this activity nationwide through the CBOs); Deploying a mobile connectivity solution using Unstructured Supplementary Service Data (USSD) to reduce turnaround time for baseline investigations results; Verification of reported pre-treatment deaths using verbal autopsy; Decentralization of treatment initiation to the local government levels; and employing a (volunteer) liaison officer (VLO) for state-wide coordination of activities. Series of trainings for capacity building was conducted for all cadres of frontline health workers who implemented the interventions. This was followed by close, supportive supervision and mentoring throughout the implementation process.

**Study sample.** We used maximum variation purposive sampling to ensure that our sample included a diverse range of individuals with different backgrounds, roles, and experiences related to the decentralization of DRTB treatment [10]. Akwa-Ibom and Oyo States have 31 and 33 Local Government Areas (LGAs) respectively. A binary categorization of good vs. poor implementers was applied to each LGA based on the proportion of DR-TB patients enrolled during the 15-months implementation period, as well as the time to enrollment. Then, LGAs were further stratified into high vs. low burden stratum based on their DR-TB case notification rate (per 10,000). Thus, four categories consisting of Low Burden Good Performance; Low Burden Poor Performance; High Burden Good Performance; and High Burden Poor Performance LGAs emerged. We interviewed participants (frontline healthcare workers) within two LGAs drawn from each of these categories (total of 16 LGAs), and additionally purposively selected stakeholders from the State TB Control Program who were directly involved in implementing the interventions.

## Participants recruitment

One-on-one interviews were conducted with diverse stakeholders who were directly involved in implementing the project interventions in both states. These stakeholders were purposively selected as key informants between July 10, 2023 and August 11, 2023, allowing variability among all relevant stakeholders considering age, sex, years of experience and different cadres. We included only health workers who had worked within at least 1-year of the project intervention implementation. The participants were selected from both state and LGA level program staff which included the following personnel; two state program managers, two state DR-TB focal persons, two state (lab) quality assurance officers, two state logistics officers, two

officers from community-based organizations, two clinicians, four DR-TB survivors, sixteen TBLSs, four DOTS providers and four managers of private laboratories. In all, a total of 40 key informant interviews (KIIs) were conducted.

**Data collection.** The CFIR Interview Guide tool was used to inform our semi-structured interview guide [11]. (S1 Appendix) The interview guide included two sections; the first on general enquiries about their personal opinions on the project interventions; and the second section on questions within the five CFIR domains and items relevant to the study and/or the implemented interventions. KIIs were conducted in-person, by the two trained qualitative researchers in August 2023. Data collection was conducted until data saturation was achieved, resulting in a total of 40 key informant interviews (KIIs). All interviews were audio-recorded, lasted for about 40–50 minutes and all the selected personnel participated in the study. There were no repeat interviews for any of the participants. The KIIs were conducted in the individual's offices at their own convenient time in English language. All audio recordings from the interviews were securely stored on password-protected in computers with restricted access, and backup copies were encrypted and stored on secure external drives. Afterwards, all the recordings were professionally transcribed verbatim. The transcripts were not returned to the participants for correction but the result of the study was presented to a group of stakeholders which included some participants in the study.

**Data analysis.** A template analysis of interview transcripts was done to identify themes describing barriers and enablers to implementing the project's interventions in relation to the CFIR constructs. We adopted the CFIR constructs as our coding template, using a deductive coding approach, rather than generating a coding template with our subset of data. The CFIR constructs and sub-constructs were identified as a priori codes for an initial codebook. Three researchers, ENO, CA and NMO participated in refining the codebook to operationalize the definitions of each domain for the project. Once consensus was achieved, two researchers, ENO and CA coded the transcripts independently with discrepancies resolved through discussion. The team met after all the transcripts were coded to discuss preliminary themes and to reach a consensus using the updated CFIR template as a guide. Transcripts were anonymized and imported into NVivo 12 software for coding and analysis.

**Ethical consideration.** Ethical approval was granted by the National Health Research Ethics Committee, Nigeria NHREC/01/01/2007-15/06/2023. Written informed consent was obtained from all individual participants included in the study. Additionally, all participants gave their consent for publication of their anonymised interview transcripts.

## Results

Study findings were reported in line with the Standards for Reporting Qualitative Research (SRQR) checklist (S2 SRQR Checklist in S2 Appendix) [12]

### Interviewers' characteristics

The authors ENO and CA conducted the interviews. Both are fellows of the West African College of Physicians (WACP), ENO is a Senior Lecturer at Ebonyi State University Abakaliki, Nigeria while CA is a Consultant Public Health Physician at Alex-Ekwueme Federal University Hospital, Abakaliki, Nigeria. Both are males and have some experience in qualitative studies [13–16].

### Participants' profile

The age of the respondents ranged from 23 to 59 years. Most of the participants, 65% were females. Half of the respondents have attained tertiary education. For those in the services of the National TB Control Program, more than half have served for 5 years and above. About

one fifth of the participants have been in the services of the National TB Control Program for more than 10 years.

**Participant's impression about the interventions.** Overall, individual opinions about the project intervention components or "thing" being implemented varied (see details under innovation design below). However, all the participants acknowledged the positive impact of the project with the impression that it made significant contributions to reduce pre-treatment loss to follow-up, prompt initiation of treatment, as well as positive impact on healthcare workers. Some respondents expressed their opinion as follows:

> *'Initially, we used to have a bulk number of our patients who were diagnosed with DR-TB, I mean DR-TB patients who are diagnosed but are not on treatment due to a number of reasons, such as: this one is not living here; he has changed address; he died before the result came out; and a number of excuses, but this project has really helped to stop that.' (B014)*

> *'And according to the project's guidelines, a patient should be tracked for treatment within 10 days of diagnosis. So that has brought about a positive change in healthcare workers attitude and behavior about prompt identification and treatment of persons diagnosed with DR-TB, as we strive to keep to the timeline.' (B02)*

> *'One of the interventions that I've seen as a welcome development in the wave 9 project is when they used a DR-TB survivor to also assist the newly diagnosed cases in basic training. It also helped in encouraging the patients to adhere to their treatment. I've realized that before now you see the State Program Officers threaten patients with police so as to take their drugs. That is different this time around. So, whoever brought the idea of using DR-TB survivors to join the counseling I say a big thank you to that person.' (A05)*

**Identified implementation barriers and enablers by CFIR domains.** Fig 1 shows the coding tree for this qualitative study with constructs that were identified from the data analysis in colored boxes.

Thirty-eight themes emerged from the interviews, which were categorized into 15 CFIR constructs under the five domains (Fig 1). Majority of the emergent themes were grouped under the Innovation Characteristics (17 of 38 themes) and Inner Setting (11 of 38 themes) domains respectively. Results are presented below, organized by CFIR domains and constructs, and by identified enablers and barriers. These are summarized alongside with participants' quotes in the supporting file, S1 Table.

**Domain:**

*Domain 1: Innovation*. Within the innovation domain, constructs that emerged were Innovation Relative Advantage, Innovation Complexity, Innovation Design and Innovation Cost. Themes of identified enablers and barriers are summarized as follows:

A. **Innovation Relative Advantage**

*Enabler:* Most of the participants attested that the interventions to decentralize services brought about faster notification of newly diagnosed DR-TB patients, brought treatment closer to the people, enhanced patient tracking, improved ease of conducting baseline investigations, reduced transportation barrier thus improving access to DR-TB treatment services, empowered health workers, and improved counseling using DR-TB survivors.

Participants appreciated the WhatsApp group for quickly alerting them to new DR-TB cases, enabling prompt treatment initiation. Decentralized pre-qualified labs brought treatment services closer, reducing patient travel distance, costs, and time. Sharing

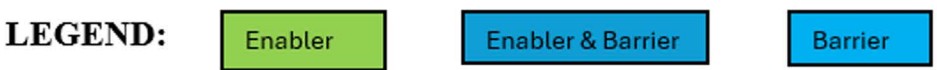

Figure 1: Coding tree highlighting implementation enablers and barriers adapted from the Consolidated Framework of Implementation Research (CFIR)|

**Fig 1. Coding tree highlighting implementation enablers and barriers adapted from the Consolidated Framework of Implementation Research (CFIR).**

lived-experience by DR-TB survivors reportedly improved pre-treatment counselling and was lauded by several respondents.

B. **Innovation Complexity**

*Enabler:* The intervention was not complicated in any form.

TBLSs (who were the stakeholder group most directly involved in interventions implementation as they traditionally oversee community-level DR-TB care), felt decentralization was not complex, nor did it change their primary scope of DR-TB work with diagnosis, conducting baseline tests, and treatment duties.

C.  **Innovation Design**

*Enabler:* Bundled intervention packages that the participants applauded most were: unique role of the VLO in facilitating prompt treatment initiation, prompt DR-TB diagnosis alert from the WhatsApp group, improved patient tracking, involvement of the private sector, and the use of DR-TB counselors which a participant dubbed a game changer.

Several participants described the VLO's role as crucial for prompt DR-TB enrollment by coordinating relevant health personnel to promptly commence activities for patient tracking, baseline tests, and pre-treatment counseling sessions where a DR-TB survivor shared their lived-experience.

*Barrier:* Interventions identified as less useful to decentralization were verbal autopsy, which was perceived as stigmatizing by the relatives, and USSD which was not considered as a priority for the program.

Participants believed obtaining information about dead patients using verbal autopsy was perceived as stigmatizing as inquiring about the circumstances of a patient's death made patients' relatives feel stigmatized. Overall, slow uptake of USSD by the end-users (particularly TBLS and the Lab FP) was the major reason for its underutilization thus, employing mobile connectivity solutions to improve turnaround time of baseline investigations using the technology was considered unimportant.

D.  **Innovation Cost**

*Barrier:* A theme that emerged within this construct is pricing of baseline tests.

All the lab FPs complained that the Government-approved (NTBLCP-Global Fund) pricing for baseline investigations was far less than the market price, and thus a huge barrier for involvement of the private sector. In one of the zones, there was (temporarily) no functional peripheral lab at some point during the project implementation because the engaged laboratory pulled out of the agreement for this reason.

*Domain 2: Outer setting.* The only CFIR construct identified within the outer setting domain was critical incidents.

A.  **Critical incidents**

*Enabler:* Unanticipated events inadvertently justified the need for decentralization such as the removal of fuel subsidy by the Federal Government of Nigeria which triggered an increase in transport fares. Furthermore, the prevailing economic difficulties resulted in increased cost of living for an already economically disadvantaged group.

*Barrier:* Decentralization efforts were grossly limited by unavailability of suitably qualified lab facilities to provide comprehensive services in some rural areas. Another key issue identified as an implementation barrier was poor power supply which heightened operational costs.

*Domain 3: Inner setting.* The constructs (and sub-constructs) identified under the inner setting domain include Structural characteristics (Physical Infrastructure and Information Technology Infrastructure), Culture, Tension for Change, Compatibility, Available Resources (Materials & Equipment), and Access to Knowledge & Information. The following are themes of enablers or barriers to the implementation efforts:

*Enabler:* Many participants identified existing physical infrastructure (DOTS centers) and a dedicated health workforce as key enablers. Decentralization was timely, addressing issues with tension for change like treatment refusal by patient due to distance, attendant transport and/or opportunity costs, and lab delays. The interventions aligned with daily work, program objectives, and provided necessary resources and training for implementation.

*Barrier:* An identified barrier was the need for more DR-TB expertise in peripheral areas.

Decentralization of DR-TB services meant shifting care beyond the state capital to rural areas which have less trained personnel with expertise on DR-TB management.

*Domain 4: Individuals*. Within the individuals' domain, the constructs identified are capability, opportunity and motivation.

*Enabler:* Participants highlighted confidence, self-efficacy, passion and available opportunities for intervention implementation as enablers.

Health workers expressed confidence and a sense of self-efficacy to carry out the interventions. This stemmed from acquired competence from repeated trainings (for both the intervention as well as on-the-job), gaining requisite skills to fulfill their role. Embedding interventions within routine programming and practice created the opportunities for implementation. Most participants expressed a high degree of personal commitment and intrinsic motivation from the satisfaction of seeing DR-TB patients recover.

*Barrier*: Notable factors identified as barriers include: the fear of being infected with TB among some health workers, as well as a reported sense of apathy among other healthcare workers.

There was fear of being infected with TB among health workers leading to self-protective behaviours such as avoiding close contact with DR-TB patients.

*Domain 5: Process*. Within the process domain, the identified construct was planning.

A. **Planning**

*Enabler:* Emerging themes under this construct were outlining specific steps and workspace, and appointment of desk officers.

To aid the implementation process, practical strategies undertaken by various stakeholders included outlining specific procedures, workspaces and assigning roles and responsibilities. For instance, structured checklists and job aids were deployed as needed by the intervention implementers. With regards to planning, traditional project management preparatory activities and measures were put in place such as having project-defined goals and tracking specific indicators.

## Discussion

This study aimed to identify enablers and barriers that shaped the implementation of DR-TB decentralization services in two states of southern Nigeria by applying the CFIR to provide actionable insights for programmatic scale-up. Based on their respective implementation experiences, stakeholders who mainly consist of State and LGA-level TB program managers, clinicians, and frontline health workers (DOTS FPs, and lab FPs) shared their respective implementation experiences, and described factors related to all five CFIR domains (intervention characteristics, outer setting, inner setting, characteristics of individuals and process).

### Contextual factors influencing decentralization of DR-TB services by CFIR domains

Overall, participants' impression about the impact of the entire decentralization initiative was positive. We identified 30 enablers and 8 barriers under 15 constructs of the five CFIR domains. Most of the identified constructs were within the Innovation (*Relative Advantage, Innovation Complexity, Innovation Design, Innovation Cost*) and Inner Setting (*Structural characteristics, Culture, Tension for Change, Compatibility, Available Resources, and Access to Knowledge & Information*) domains. *Critical incidents* and *Planning* were the only Outer Setting and Implementation Process dynamics identified within the respective domains; while

the notable behavioural constructs describing characteristics of Individuals identified were *Capability, Opportunity, and Motivation.*

Our findings showed that most of the contextual factors perceived to influence the implementation of decentralized DR-TB services were within the Innovation and Inner Setting domains of the CFIR. Participants' feedback on perceived advantages of the decentralization intervention correlate with findings from a systematic review of factors influencing MDR-TB diagnosis and treatment initiation which highlighted decentralization of services among others as a facilitator, and hence recommended decentralization [17]. Furthermore, within the inner setting domain, identified sub-constructs are inclusive of both those that exists regardless of implementing any interventions (e.g., structural characteristics like existing physical infrastructure and/or the health workforce culture), as well as factors that are specific to the interventions' implementation, e.g., tension for change, compatibility, available resources, and access to knowledge & information. The abundance of these contextual factors suggests flexible opportunities that could be targeted by implementation strategies for interventions scale up.

## Enablers to decentralization implementation

Many of the identified themes classified as enablers are within the innovation domain, and the relative advantages of the decentralized approach as indicated by stakeholders are: faster notification of newly diagnosed DR-TB patients, empowered healthcare workers, enhanced patient tracking, improved ease of conducting baseline investigations, brought treatment closer to the people, reduced transportation barrier thus improving access to DR-TB treatment services, and improved counseling using DR-TB survivors. These findings are consistent with similar evidence of improved DR-TB services and is also in keeping with the calls for more decentralization of TB services globally [5,18,19].

Participants also emphasized its simplicity, and the well-packaged design of the interventions. Despite being a multilevel mix of interventions, some implementation strategies may have had more impact on implementing the decentralized approach. Those highlighted by stakeholders are the: i). Use of WhatsApp platform for result notification; ii). Coordinating role of the VLO; iii). Use of DR-TB survivors as peer counselors; iv). Engagement of the private sector (for laboratory services); v). Improved patient tracking by CBOs. While some of these identified themes were novel, most of them corroborated findings in the literature as previous studies have reported implementation success by applying similar strategies. For instance, inclusion of DR-TB survivors (dubbed a 'game changer' by a respondent) for team counselling to reduce treatment refusal and aid prompt treatment initiation and adherence, as well as stigma reduction was well received. This was the experience of other studies that included survivors of other diseases as counselors resulting in reduced loss to follow-up [20,21].

Decentralizing laboratory services for the conduct of baseline investigations through private sector engagement improved person-centered service delivery by significantly reducing travel distance, time and cost as well as effectively bringing DR-TB services closer to the communities. Multi-country experiences have generated abundant evidence that private sector engagement is invaluable for effective TB control through for-profit involvement across the spectrum of TB programming, including laboratory services [22–24].

The use of digital tools such as the state-wide WhatsApp group for prompt DR-TB result notification enabled frontline health workers to obtain patients' results earlier than the hardcopy. This triggered necessary action for relevant activities for treatment initiation thereby, aiding accountability for every patient. Thus, the use of the WhatsApp group positively changed the attitude of the healthcare workers by stimulating a shorter response time for

treatment initiation. This is not surprising as in a similar setting in Tanzania, WhatsApp was used as a revised communication strategy to reduce lab turnaround time and resulted in 22% and 36% reduction in the times for specimen transportation from health facilities to the lab, and median time for the district to receive results respectively [25]. Use of ubiquitous WhatsApp was an appropriately fit technology for the study setting, and unlike the DR-TB IMS platform/USSD, there was no learning curve associated with use of WhatsApp groups.

Key enablers identified within the inner setting included the ability to leverage the existing health system infrastructure such as the functional DOTS centres and staff. Implementing interventions under routine programmatic conditions coupled with the adaptability of the interventions to align with the usual workflow was instrumental for compatibility, and invariably eased implementation. There is evidence that if a program does not fit with the norms and practices already existing in the health facility, the implementation process may be hampered [26]. Tension for change which reflected as readily identifiable gaps within the default DR-TB component of the program by participants highlighting challenges with patient tracking, travel time to conduct baseline investigations, treatment refusals, and overall poorer outcomes was already indicative of, and could have contributed to a readiness for implementation as has been seen in other conditions implementing evidence-based practices (EBP) [27]. Trainings and capacity building with access to knowledge & information, as well as available resources were noted to aid implementation. Identifying these inner setting dynamics is important to understand the local implementation experience, and they may be modifiable drivers of implementing evidence-based practices [28]. Furthermore, themes identified are indicative of health systems strengthening at sub-national levels as several building blocks were impacted by the multilevel interventions, and this is commendable for sustainability as multiple country experiences have shown advances in TB control and system strengthening are complementary [29].

The role of broader systems and spaces including but not limited to fiscal policy and geopolitical factors in influencing implementation, scalability, and sustainability of EBP is well established and are arguably more critical in resource-limited settings such as Nigeria [7,30]. In this study, most of the outer setting factors identified were barriers, however, a paradoxical implementation enabler was the policy move that mandated the sudden removal of subsidy on petroleum products by the Federal Government of Nigeria in May 2023 which brought economic hardship from the attendant hike in transportation prices and cost of living. Suffice it to say that the impact of fuel subsidy removal in Nigeria has made it imperative, if not mandatory to further decentralize DR-TB services in-country. TB is associated with poverty and transportation cost is implicated in contributing to catastrophic costs for households affected by TB. Interventions and policies to reduce indirect TB costs such as decentralized services (especially laboratory and treatment initiation at communities) must then be actively encouraged in line with the global End TB targets to alleviate suffering for those affected by the disease.

Individual actions and behaviour influence effective project implementation which reflects the extent of achieving the patient's needs [6]. In describing the individual characteristics domain, only the concept of 'need' was not identified as a relevant factor. Participants felt that they were physically and psychologically capable of implementing the intervention, had the opportunities to do so, and were motivated to implement the interventions. Self-efficacy of the participants was linked to the frequent training and re-training of the health workers. This observation highlights the importance of regular training of health care workers and supportive supervision as they perform their duties to improve their knowledge and confidence for successful implementation of an initiative [31–33]. Participants were highly motivated to see their patients recover from ill health and prevent the spread of the disease in the interest of protecting public health. This is commendable

and could be the effect of a sustained drive and commitment to find the missing millions through consistent active case-finding initiatives that has been the hallmark of the NTBLCP strategic plans since the past decade [34].

Effective planning was the only facilitator at the process level and themes identified were assignment of roles and responsibilities, as well as outlining intervention-specific steps and workspace in advance. Certain tasks were done with structured checklists as job aids, and there were project-defined goals with specific indicators monitored.

### Barriers to decentralization implementation

Overall participants' feedback specific to the multicomponent intervention revealed that verbal autopsy and USSD interventions were the least implemented and some outrightly opined the interventions should either be modified or dropped entirely. Fear of stigmatization by the dead patients' relatives was a major barrier that limited the use of verbal autopsy as planned and while this may be unconnected to cultural sensitivity to discussing the dead in Africa, it certainly underscores the reality of TB-related stigma in the communities [35–37]. Overall, the uptake and use of USSD by end-users (TBLS and Lab FP) was slow. While this may be partly explained by diverse reasons for late adoption of digital interventions among frontline healthcare workers [38,39], this technological intervention was considered unimportant by the participants. Also, within the innovation domain, low pricing of baseline tests below the market price was a huge barrier identified given that expansion of laboratory services mainly entailed engaging the private sector. This finding is unsurprising as evidence for sustained program partnership with the private sector requires substantial and continuous support, including financial support [40].

Poor power supply and inadequacy of laboratory facilities in rural areas are the core identified barriers external to the TB program implementation setting. While the use of alternative power supply significantly raises operational costs, decentralization efforts were increasingly hindered by unavailability of suitably qualified lab facilities towards more rural settings, and potentially limits further decentralization of services. Understandably, DR-TB expertise diminishes away from the city-centre with less competent hands in rural areas necessitating the need for more capacity building.

Fear of being infected with TB among health workers has been reported severally and directly linked to their service delivery [41,42], thus, an important barrier to be addressed to optimize decentralization of DR-TB services.

Decentralization of DR-TB services is notably cost-effective especially in low-resource settings; however, it could be very challenging and requires varying mix of interventions to address local operational challenges [5]. In this study, we identified more facilitators, and the key enablers centered around the intervention characteristics, individual characteristics, and features of the inner setting within the TB program implementation space. Conversely, majority of the identified barriers were from two intervention strategies that seemed ill-suited, and from outer setting dynamics of fiscal policy and geographic access barriers that were external to the program. While the global push towards more person-centered TB services makes decentralization more imperative, implementation strategies must be adaptable to the local context. In addition to documenting intervention successes, reporting implementation experiences and outcomes with the use of a standardized framework like the CFIR is helpful for reporting lessons learnt to glean pragmatic insights and evidence to guide policy and practice.

### Strengths and limitations

This project piloted the decentralization of DR-TB treatment enrollment services in Nigeria with a systematic evaluation of its implementation within routine national TB

programming. Being a multilevel complex health intervention, using a determinant's framework as the CFIR contextualizes study findings, facilitates comparison, and contributes to global implementation science research. Our study has some limitations. The use of digital health interventions has potential pitfalls such as confidentiality issues as WhatsApp does not have mandatory password protection, and images maybe locally saved onto the user's photo library on their phone. However, in this study, the free tool was primarily used to enhance communication by reducing the turnaround time for retrieving lab result diagnosis of DR-TB, and we relied on WhatsApp's end-to-end encryption feature for data protection.

Selection bias was caused by using a purposive sample from the top 10 states with the highest burden of PTLTFU thus, results could differ in some states. The two study states are in the southern part of the country, and being in different geo-political zones, notwithstanding, they have comparable data with many other states in the country and share similar problems with regards to DR-TB service delivery. Consequently, the findings from this study though not generalizable, will be useful for scaling up and sustaining decentralization of DR-TB services in Nigeria.

## Conclusion

Findings provide strong evidence supporting the ease of implementing DR-TB decentralization services and highlight its relative advantages over the centralized approach. Key implementation enablers centered around the innovation characteristics, individuals, and the inner setting dynamics within the TB program. Despite some external barriers relating to fiscal policy and geographic access, there were more facilitators than barriers identified among the key contextual factors influencing intervention implementation, most of which are modifiable. Applying the CFIR for a post-implementation evaluation was apt for defining contextual factors which is useful for comparison and for a common language in global implementation science research. Insight gained can be applied to facilitate nationwide adoption and scale-up of decentralized DR-TB services in Nigeria, as well as similar settings in low-and-middle-income-countries.

## Supporting information

**S1 Appendix.  Key Informant Interview Guide.**
(DOCX)

**S1 Table.  Above shows the quotes for all the domains.**
(DOCX)

**S2 Appendix.  Standards for Reporting Qualitative Research (SRQR).**
(DOC)

**S3 File.  KII/A1.**
(DOCX)

## Acknowledgement

The authors would like to thank the Akwa-Ibom and Oyo States' Tuberculosis and Leprosy Control Program (STBLCP) teams, particularly their Program Managers and DR-TB focal persons (Dr. Akpan Bassey and Mrs. Magdalene Abasiekwere in Akwa-Ibom), and (Dr. Johnson Babalola and Mrs. Esther Oyebamiji in Oyo) respectively, for their immense support throughout the TB Reach Wave 9 project implementation.

## Author contributions

**Conceptualization:** Ngozi Murphy-Okpala, Chinwe Eze.

**Data curation:** Edmund Ndudi Ossai, Chibuike Innocent Agu.

**Formal analysis:** Edmund Ndudi Ossai, Chibuike Innocent Agu, Sode Matiku, Beatrice Kirubi, Jacob Creswell, Joseph Chukwu.

**Funding acquisition:** Ngozi Murphy-Okpala.

**Investigation:** Ngozi Murphy-Okpala, Chinwe Eze, Edmund Ndudi Ossai, Chibuike Innocent Agu, Ngozi Ekeke, Anthony Meka, Sode Matiku, Martin Njoku, Victor Babawale.

**Methodology:** Ngozi Murphy-Okpala, Chinwe Eze, Edmund Ndudi Ossai, Ifeyinwa Ezenwosu, Ngozi Ekeke.

**Project administration:** Ngozi Murphy-Okpala, Chinwe Eze, Charles Nwafor, Joseph Chukwu.

**Resources:** Ngozi Murphy-Okpala, Charles Nwafor, Beatrice Kirubi, Joseph Chukwu.

**Software:** Ngozi Murphy-Okpala, Chinwe Eze, Sode Matiku, Francis S. Iyama.

**Supervision:** Ngozi Murphy-Okpala, Chinwe Eze, Charles Nwafor, Anthony Meka, Okechukwu Ezeakile, Francis S. Iyama, Jacob Creswell, Victor Babawale, Chukwuma Anyaike, Joseph Chukwu.

**Validation:** Ngozi Murphy-Okpala, Chinwe Eze, Edmund Ndudi Ossai, Charles Nwafor, Ngozi Ekeke, Anthony Meka, Sode Matiku, Beatrice Kirubi, Okechukwu Ezeakile, Martin Njoku, Francis S. Iyama, Jacob Creswell, Victor Babawale, Chukwuma Anyaike, Joseph Chukwu.

**Visualization:** Ngozi Murphy-Okpala, Chinwe Eze, Edmund Ndudi Ossai, Charles Nwafor, Ngozi Ekeke, Anthony Meka, Sode Matiku, Beatrice Kirubi, Okechukwu Ezeakile, Martin Njoku, Francis S. Iyama, Jacob Creswell, Chukwuma Anyaike, Joseph Chukwu.

**Writing – original draft:** Chibuike Innocent Agu, Ifeyinwa Ezenwosu.

**Writing – review & editing:** Ngozi Murphy-Okpala, Chinwe Eze, Edmund Ndudi Ossai, Chibuike Innocent Agu, Ifeyinwa Ezenwosu, Charles Nwafor, Ngozi Ekeke, Anthony Meka, Sode Matiku, Beatrice Kirubi, Okechukwu Ezeakile, Martin Njoku, Francis S. Iyama, Jacob Creswell, Victor Babawale, Chukwuma Anyaike, Joseph Chukwu.

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
