## [Decision Letter · Decision Letter 0]

3 Sep 2024

PONE-D-24-27141Enhancing Programmatic Scale-Up: Applying the Consolidated Framework for Implementation Research to Evaluate Decentralized Drug-Resistant Tuberculosis Services in Southern NigeriaPLOS ONE

Dear Dr. Agu,

Thank you for submitting your manuscript to PLOS ONE. After careful consideration, we feel that it has merit but does not fully meet PLOS ONE’s publication criteria as it currently stands. Therefore, we invite you to submit a revised version of the manuscript that addresses the points raised during the review process.

We look forward to receiving your revised manuscript.

Kind regards,

Pengpeng Ye

Academic Editor

PLOS ONE

**Journal Requirements:**

The research leading to these results was funded by the Stop TB Partnership under TB REACH Wave 9 grant number: STBP/TBREACH/GSA/W9-9833. However, the views expressed do not necessarily reflect the views of Stop TB Partnership, but belong solely to the authors.

Reviewers' comments:

Reviewer's Responses to Questions

**Comments to the Author**

1. Is the manuscript technically sound, and do the data support the conclusions?

Reviewer #1: Yes

Reviewer #2: Yes

2. Has the statistical analysis been performed appropriately and rigorously? 

Reviewer #1: Yes

Reviewer #2: Yes

3. Have the authors made all data underlying the findings in their manuscript fully available?

Reviewer #1: Yes

Reviewer #2: No

4. Is the manuscript presented in an intelligible fashion and written in standard English?

Reviewer #1: Yes

Reviewer #2: Yes

5. Review Comments to the Author

**Reviewer #1: ** The project piloted the decentralization of DR-TB treatment registration services in Nigeria, and systematically assessed its implementation in national TB programming. The study use CFIR as framework to evaluate the facillitators and barriers of the intervention and contributes to implementation science research. The study decription was quite clear and facillitators and barriers of the intervention have been identified. There are some questions regarding the research.

Methods：

1. (line 185) Please describe criteria for deciding when no further sampling was necessary in the data collection part (e.g., sampling saturation).

2. Please describe data management and security, anonymization/deidentification of excerpts.

Results：

1. (line 251-252) Majority of the emergent themes were grouped under the Innovation Characteristics (17 of 38 themes) and Inner Setting (11 of 38 themes) domains respectively. Can you explain the possible reasons why the remaining three parts have less content identified?

2. In the CFIR analysis results section, some excerpts can be added to help understand the content and increase credibility.

**Reviewer #2: ** Summary: This is a retrospective analysis of a pilot study testing the decentralized approach to drug-resistant TB services in two states in Nigeria. The authors conducted and qualitatively analyzed interviews with key informants using the Consolidated Framework for Implementation Research (CFIR). The authors do a nice job of applying the CFIR in a way that is methodologically sound and well explained. The findings show a good number of enablers to implementing this approach and describe barriers to implementation; the work is useful for informing further adoption and scale-up of such an approach.

Overall comments

• Implementation science application: There is a lot of really rich detail that is well-laid out and clear to follow because of the application of the CFIR. It would help clarify a great deal if the language in the intervention description aligned directly with what is presented in the results so that it can be referred back to.

o It is difficult to follow what is the intervention and what are the implementation strategies to deliver the intervention.

o For example, in the discussion, the authors describe implementation strategies that were previously mentioned as part of the innovation (e.g. WhatsApp) – please clarify what is the intervention and what are implementation strategies

o Line 269 – first mention of the WhatsApp group – is this part of the innovation? Please add to the intervention description or clarify particularly as it a large part of the discussion

o Some inconsistent use of the CFIR and the CFIR domain/construct terms use consistent CFIR terms (e.g. external should be outer setting?)

o Line 118, 363, 529 - Don’t need to say “CFIR framework” as “framework” is already in the name

o COM-B is introduced in the discussion – was this model used to guide analysis and interpretation of the findings? It needs to be presented in the methods and the results rather than only the discussion if it helped to guide the work. Furthermore, COM-B itself is not well-explained on what it is, what the components are, and why it is applied. This explanation of the model belongs in methods.

• The authors may need clarification on what availability of data means. This would include access to all fully transcribed interviews, which I don’t believe are included in full within the manuscript. Currently, the statement reads, “All relevant data are within the manuscript and its Supporting information files”

Minor comments

• Line 164 – additional explanation of what maximum variation purposive sampling is (with citation) and why it was used would be helpful

• Line 175, is there a word or phrase missing? One-on-one interviews with diverse stakeholders who were directly involved in implementing the project interventions in both states.

• Line 186 – capitalization of “CFIR Interview Guide” tool suggests it came from a specific place (CFIRguide.org?) - please cite where the guide comes from

• Line 199 – does this mean you undertook a deductive coding approach rather than inductive? Please clarify this sentence, “We adopted the CFIR constructs as our coding template rather than generating a coding template with our subset of data.”

• Line 271 – unclear how “sharing lived experience” fits in relative advantage? Can you clarify/explain more clearly what about the intervention design compared to the centralized approach makes it possible to share lived experience?

6. PLOS authors have the option to publish the peer review history of their article (what does this mean? ). If published, this will include your full peer review and any attached files.

**Do you want your identity to be public for this peer review?** For information about this choice, including consent withdrawal, please see our Privacy Policy .

Reviewer #1: **Yes: ** YE JIN

Reviewer #2: **Yes: ** Natalie A Blackburn

---

## [Author Response · Author response to Decision Letter 1]

24 Dec 2024

Academic Editors’ comments

Comment: Please state what role the funders took in the study.

Response: This has been stated as, “The funders had no role in study design, data collection and analysis, decision to publish, or preparation of the manuscript" in the manuscript.

Comment: If your ethics statement is written in any section besides the Methods, please move it…

Response: The ethics statement has been moved to the methods section

Comment: Please include captions for your Supporting Information files at the end of your manuscript

Response: This has been done

Comment: Please review your reference list to ensure that it is complete and correct. If you have cited papers that have been retracted

Response: Done. Article no 29 in the previous reference list was removed as it is no longer relevant following revisions, while article no. 10 and 11 in the current list have been newly added.

Comment: Please confirm at this time whether or not your submission contains all raw data required to replicate the results of your study.

Response: We can confirm that all raw data required to replicate the results of our study have been submitted. The data availability statement has been revised to ‘All relevant data are available in the Supporting Information file (S3 TRANSCRIPT)’

Comment: Can you please confirm that all participants gave consent for interview transcript to be published?

Response: We can confirm that all gave consent for interview the anonymised interview transcript to be published. This has been stated in the ethical consideration sub-section of the manuscript

Comment: please also confirm that the transcripts do not contain any potentially identifying information

Response: We can confirm that the transcripts do not contain any potentially identifying information. Thank you.

Reviewer #1

Comment: Please describe criteria for deciding when no further sampling was necessary in the data collection part (e.g., sampling saturation):

Response: Thank you for your comment. This has now been described in the data collection sub-section (page 9)

Comment: Please describe data management and security, anonymization/deidentification of excerpts

Response: these have been described in the methods (page 9).

Comment: Majority of the emergent themes were grouped under the Innovation Characteristics (17 of 38 themes) and Inner Setting (11 of 38 themes) domains respectively. Can you explain the possible reasons why the remaining three parts have less content identified?

Response: Outer Setting, Individual Characteristics, and Process may have received less attention due to the participants' roles, or the stage of implementation, where external or outer setting factors, individual-level issues, or specific implementation strategies may not yet be fully prominent. On the other hand, Innovation Characteristics and Inner Setting likely reflects their central role in the decentralization of DRTB treatment, and are often the most immediate concerns for key informants

Comment: In the CFIR analysis results section, some excerpts can be added to help understand the content and increase credibility.

Response: Thanks for your observation. All quotes were summarized in Table 1 to maintain a manageable word count. Some excerpts have now been included accordingly.

Reviewer #2

Comment: o It is difficult to follow what is the intervention and what are the implementation strategies to deliver the intervention…

Response: Thank you for this observation. The core evidence-based intervention was Decentralization of DR-TB treatment initiation services which we implemented as a bundle of interventions that had to be deployed at multiple levels along the patient care pathway, as listed on page 7 under the ‘description of the interventions’. Additional explanation have been included for better description. This paper presents results from the evaluation of the implementation experience during a pilot project to decentralize DRTB treatment initiation (lines 119-120 of the manuscript). As such, the focus of this study was to assess the implementation outcomes of feasibility, acceptability, and understand contextual factors influencing the implementation of the decentralized model.

Comment: …first mention of the WhatsApp group – is this part of the innovation? Please add to the intervention description or clarify particularly as it a large part of the discussion

Response: Thank you for the observation. This has been corrected with a more detailed description of the intervention section, including the role of WhatsApp as with other components of the intervention mentioned in the ‘description of intervention’ sub-section of the methods section (page 7)

Comment: Some inconsistent use of the CFIR and the CFIR domain/construct terms use consistent CFIR terms (e.g. external should be outer setting?)

Response: Thank you for your clarification. The correction has been made, and the term "external" has been replaced with "outer setting" in the appropriate context. However, in instances where "external" was not specifically referring to the outer setting, it has been retained to maintain the intended meaning.

Comment: Don’t need to say “CFIR framework” as “framework” is already in the name

Response: This has been corrected

Comment: COM-B is introduced in the discussion – was this model used to guide analysis and interpretation of the findings? It needs to be presented in the methods and the results rather than only the discussion if it helped to guide the work…

Response: Thank you for the observation. The statement have been revised, and the COM-B model removed as it was not originally included in the analysis or interpretation.

Comment: The authors may need clarification on what availability of data means. This would include access to all fully transcribed interviews, which I don’t believe are included in full within the manuscript. Currently, the statement reads, “All relevant data are within the manuscript and its Supporting information files”

Response: All data relating to the manuscript have been included as supporting information named, ‘Transcript’

Comment: additional explanation of what maximum variation purposive sampling is (with citation) and why it was used would be helpful

Response: additional explanation along with a citation has been provided

Comment: is there a word or phrase missing? One-on-one interviews with diverse stakeholders who were directly involved in implementing the project interventions in both states.

Response: Thank you. This has been corrected

Comment: capitalization of “CFIR Interview Guide” tool suggests it came from a specific place (CFIRguide.org?) - please cite where the guide comes from

Response: Citation has provided

Comment: does this mean you undertook a deductive coding approach rather than inductive? Please clarify this sentence, “We adopted the CFIR constructs as our coding template rather than generating a coding template with our subset of data.”

Response: This has been clarified (page 9).

Comment: unclear how “sharing lived experience” fits in relative advantage? Can you clarify/explain more clearly what about the intervention design compared to the centralized approach makes it possible to share lived experience

Response: Thank you for your comment. The inclusion of DR-TB survivors to counsel new cases was a key component of the decentralized package of interventions, which distinguishes it from the centralized approach. This element of the design supports relative advantage by offering new patients the opportunity to receive peer support from individuals who have personally gone through the challenges of DRTB treatment.

---

## [Decision Letter · Decision Letter 1]

14 Jan 2025

Enhancing Programmatic Scale-Up: Applying the Consolidated Framework for Implementation Research to Evaluate Decentralized Drug-Resistant Tuberculosis Services in Southern Nigeria

PONE-D-24-27141R1

Dear Dr. Agu,

We’re pleased to inform you that your manuscript has been judged scientifically suitable for publication and will be formally accepted for publication once it meets all outstanding technical requirements.

Kind regards,

Pengpeng Ye

Academic Editor

PLOS ONE

Additional Editor Comments (optional):

Reviewers' comments:

Reviewer's Responses to Questions

**Comments to the Author**

1. If the authors have adequately addressed your comments raised in a previous round of review and you feel that this manuscript is now acceptable for publication, you may indicate that here to bypass the “Comments to the Author” section, enter your conflict of interest statement in the “Confidential to Editor” section, and submit your "Accept" recommendation.

Reviewer #1: All comments have been addressed

Reviewer #2: All comments have been addressed

2. Is the manuscript technically sound, and do the data support the conclusions?

Reviewer #1: Yes

Reviewer #2: Yes

3. Has the statistical analysis been performed appropriately and rigorously? 

Reviewer #1: N/A

Reviewer #2: Yes

4. Have the authors made all data underlying the findings in their manuscript fully available?

Reviewer #1: Yes

Reviewer #2: Yes

5. Is the manuscript presented in an intelligible fashion and written in standard English?

Reviewer #1: Yes

Reviewer #2: Yes

6. Review Comments to the Author

Reviewer #1: (No Response)

Reviewer #2: The authors have adequately addressed comments and this work makes a strong contribution to the field.

7. PLOS authors have the option to publish the peer review history of their article (what does this mean? ). If published, this will include your full peer review and any attached files.

**Do you want your identity to be public for this peer review?** For information about this choice, including consent withdrawal, please see our Privacy Policy .

Reviewer #1: **Yes: ** Ye Jin

Reviewer #2: **Yes: ** Natalie A Blackburn

---

## [Editor Report · Acceptance letter]

PONE-D-24-27141R1

PLOS ONE

Dear Dr. Agu,

I'm pleased to inform you that your manuscript has been deemed suitable for publication in PLOS ONE. Congratulations! Your manuscript is now being handed over to our production team.

Kind regards,

on behalf of

Dr. Pengpeng Ye

Academic Editor

PLOS ONE